# Nanoproteomic Approach for Isolation and Identification of Potential Biomarkers in Human Urine from Adults with Normal Weight, Overweight and Obesity

**DOI:** 10.3390/molecules26061803

**Published:** 2021-03-23

**Authors:** Sergio G. Hernandez-Leon, Jose Andre-i Sarabia Sainz, Gabriela Ramos-Clamont Montfort, José Ángel Huerta-Ocampo, Martha Nydia Ballesteros, Ana M. Guzman-Partida, María del Refugio Robles-Burgueño, Luz Vazquez-Moreno

**Affiliations:** 1Centro de Investigación en Alimentación y Desarrollo A.C., Carretera Gustavo Enrique Astiazarán Rosas, No. 46, col. La Victoria, Hermosillo, Sonora 83304, Mexico; gerardo.hernandez@estudiantes.ciad.mx (S.G.H.-L.); gramos@ciad.mx (G.R.-C.M.); nydia@ciad.mx (M.N.B.); gupa@ciad.mx (A.M.G.-P.); cuquis@ciad.mx (M.d.R.R.-B.); 2Departamento de Investigación en Física, Universidad de Sonora, Blvd. Luis Encinas y Rosales S/N, Col. Centro, Hermosillo, Sonora 83190, Mexico; jose.sarabia@unison.mx; 3CONACyT-Centro de Investigación en Alimentación y Desarrollo A.C., Carretera Gustavo Enrique Astiazarán Rosas, No. 46, col. La Victoria, Hermosillo, Sonora 83304, Mexico; jose.huerta@ciad.mx

**Keywords:** FCSNP, Cibacron blue, LMW proteins, urinary biomarkers, HMW proteins, nanoproteomics

## Abstract

In this work, previously synthesized and characterized core-shell silica nanoparticles (FCSNP) functionalized with immobilized molecular bait, Cibacron blue, and a porous polymeric bis-acrylamide shell were incubated with pooled urine samples from adult women or men with normal weight, overweight or obesity for the isolation of potential biomarkers. A total of 30 individuals (15 woman and 15 men) were included. FCSNP allowed the capture of a variety of low molecular weight (LMW) proteins as evidenced by mass spectrometry (MS) and the exclusion of high molecular weight (HMW) proteins (>34 kDa) as demonstrated by SDS-PAGE and 2D SDS-PAGE. A total of 36 proteins were successfully identified by MS and homology database searching against the *Homo sapiens* subset of the Swiss-Prot database. Identified proteins were grouped into different clusters according to their abundance patterns. Four proteins were found only in women and five only in men, whereas 27 proteins were in urine from both genders with different abundance patterns. Based on these results, this new approach represents an alternative tool for isolation and identification of urinary biomarkers.

## 1. Introduction

Isolation and identification of low molecular weight proteins and peptides from urine is a critical need in biomedical research since they might represent a new source of biomarkers predictive of early-stage diseases [1,2]. Normal urine contains proteins, either filtered from plasma or resulting from secretion from tubular epithelial cells of segments along the nephron [3,4]. Changes in urine protein components and concentration, therefore, could report directly on dysfunction of cells within the urinary tract, while other diseases could be detectable via the filtration of analytes from blood into urine [5].

There are several reasons for considering urine an ideal medium for discovering bio- markers of diseases: urine is abundant, can be sampled easily without invasive procedures and allows for the monitoring of a wide range of physiological processes and diseases [6] as well as exposure to emerging pollutants in water [7,8]. Besides, urine has a less complex composition than serum or plasma, which reduces interferences in isolation and facilitates the evaluation of new biomarkers [9]. However, isolation and purification of biomarkers is complicated because of low concentrations that are under the detection limits of mass spectrometry and conventional immunoassays [10,11,12]. Further, these proteins or peptides could be degraded by proteases or masked by the presence of different interfering substances or high molecular weight proteins such as albumin [5,12,13]. Therefore, there is a great need to develop new tools that will selectively isolate low abundance proteins and peptides (potential biomarkers) from urine that correlate with physiological and pathological stages.

Nanoproteomics, the application of nanotechnology in proteomics [14], offers a potential means to overcome the above limitations for the isolation of biomarkers in urine. Many classes of nanoparticles such as quantum dots [15], silver and gold nanoparticles [16], hydrogels [17] and silica nanoparticles [18] have been proposed to be applicable in diagnosis, monitoring and treatment of diseases. Among these known materials, silica nanoparticles may represent a good means to capture and isolate urinary proteins or peptides of interest. Silica is a compound with appealing characteristics due to its high stability, low toxicity and, particularly, because it can be chemically modified with several compounds [19,20,21,22]. Recently, we demonstrated the ability of functionalized core-shell silica nanoparticles (FCSNP) both to capture low molecular weight proteins and peptides by the pseudo-affinity (it does not interact with proteins in a specific manner) hydrophobic molecular bait Cibacron blue, which interacts with proteins with sufficiently long hydrophobic regions, and to exclude high molecular weight proteins by the porous polymeric bis-acrylamide shell, all in a single step using model proteins [2].

In this study, to give proof of concept, we used the developed nanoparticles to separate low molecular weight (LMW) proteins and peptides from a complex source (urine) and analyzed them by single and two-dimensional sodium dodecyl sulfate-polyacrylamide gel electrophoresis (2D SDS-PAGE). Furthermore, proteins from pooled urine samples from groups of apparently healthy adults (men and women) with normal weight (control) were analyzed and identified by mass spectrometry (MS) and compared with samples from overweight and obese individuals in order to establish possible differences among them. Overweight and obesity are two conditions of critical concern to global public health and are associated with numerous chronic diseases, including cardiovascular disease, chronic kidney disease, type 2 diabetes mellitus, hypertension and hyperlipidemia [23,24].

## 2. Results and Discussion

### 2.1. Synthesis and Characterization of FCSNP

The obtained FCSNP were synthesized in four stages and characterized in our previous report [2].

### 2.2. Evaluation of the FCSNP in the Capture of LMW Proteins and Exclusion of the High Molecular Weight (HMW) Proteins

Washes with phosphate buffer were used to remove excess of proteins, while elution with different solvents was used to obtain captured proteins. In the elution with 50% isopropanol (Figure 1, lane 5), faint bands of HMW and LMW proteins were observed. This behavior was also observed in our previous report [2] and might be attributed to excess of proteins or interaction of HMW proteins with the molecular bait near the surface of the FCSNP. In elutions with 50% methanol (lane 7) and ACN+NH_4_OH (lane 9), both exclusion of HMW proteins and capture of LMW proteins was evidenced. In elution with 50% methanol, however, only one faded protein band of ~30 kDa was observed, while several LMW protein bands with a molecular mass of ~34 kDa and smaller were eluted with ACN+NH_4_OH (lane 9). This confirms that the molecular weight cutout of the polymeric shell is around 34 kDa. This is similar to the cutoff of the multifunctional core-shell hydrogels (~32 kDa) obtained by Tamburro et al. [12].

Furthermore, samples of complete urine (not incubated with FCSNP) and ACN+NH_4_OH elutions (product of the incubation of FCSNP) from a healthy individual were selected for 2D SDS-PAGE analysis. Due to the high content of salts in urine, samples from complete urine required desalting prior to 2D SDS-PAGE analysis (Figure 2a). However, in the ACN+NH_4_OH fraction (Figure 2b), removal of salts was not necessary, and only proteins under 34 kDa were observed. This confirmed that FCSNP were effective not only in the isolation of LMW proteins but also in the removal of the high salt content in urine, in a single step.

### 2.3. Evaluation of the FCSNP in the Capture of Potential Biomarkers from Pooled Urine Samples

FCSNP elutions were selected for LC-MS/MS analysis. As depicted in Figure 3, heat map offered a global view of the seven different abundance patterns (seven clusters) of all identified proteins. Proteins not detected in one group are indicated in gray. Increased protein abundance in the different groups (overweight (OW) or obesity (OB)) compared to respective normal weight (NW) conditions are indicated in green, whereas decreased protein abundances are indicated in red. Protein names are indicated on the right side of each row, while dashed blue boxes indicate protein clusters with similar abundance change patterns. A total of 36 proteins were successfully identified when at least two peptides and a score ≥25 were obtained by searching against the *Homo sapiens* subset of the Swiss-Prot database (Table 1). From the 36 total proteins, 5 were found only in men, while 4 only in women and 27 were found in both genders (Figure 4).

Furthermore, along with presence or absence, proteins were classified into seven clusters according to the protein accumulation patterns. Cluster 1 consisted in proteins found only in one gender. Of proteins found only in men: semenogelin-2 participates in the formation of the sperm coagulum [25]; prostatic acid phosphatase is produced by prostatic epithelial cells [26]; prostaglandin-H2 D-isomerase plays an important role in maturation of the central nervous system and male reproductive system [27]; ganglioside GM2 activator enables the directed movement of lipids into, out or within a cell or between cells [28] (this protein was not found in men with normal weight); and immunoglobulin, kappa variable 3D-20, the V region of the variable domain of immunoglobin light chains participates in antigen recognition [29]. The latter protein decreases in abundance as the body weight increases and might be involved in a decreased immune activity in individuals with overweight and obesity. The first three proteins (semenogelin-2, prostatic acid phosphatase and prostaglandin-H2 D-isomerase) are only produced in male organs.

Therefore, the presence of these proteins only in men’s urine indicates the reliability and accuracy of the method employed for identification. Likewise, proteins that were found only in women’s urine included vitelline membrane outer layer protein 1 homolog, which is found on the surface of the plasma membrane of an ovum [30], and hemoglobin subunit alpha, a protein usually absent from urine except in hemoglobinuria, a condition of some renal and hemolytic diseases (among others). Hemoglobin subunit alpha can also be a result of non-pathologic conditions including intense exercise or menstruation [31,32]. Interestingly, both, protein S-100-A8 and S100-9 were also found only in women. These proteins belong to the S100 family of calcium binding proteins. The S100-A8/A-9 complex induces an inflammatory response, and their expression correlates with disease severity in several inflammatory disorders [33]. Particularly in urine, the presence of these proteins reflects urinary tract infections (UTI); thus, it is probable that in the pooled samples at least one individual presented with UTI as these infections are frequent, particularly in women, affecting 12/10,000 women and 4/10,000 men annually [34].

Cluster 2 included proteins with abundance that decreases as the body weight increases. Proteins included in this cluster are involved in different pathologies. Kininogen -1 decreases in acute kidney injury as evidenced by 2D SDS-PAGE and identified by LC-MS/MS [35]; thus, this protein might be used as a potential prognostic biomarker for acute kidney injury as suggested by Gonzalez-Calero et al. [35]. In our study, kininogen-1 decreased as the body weight increased. Alpha-2-HS-glycoprotein (Fetuin-A) has been proposed as a novel marker for diabetic nephropathy in type 2 Diabetes, and it was demonstrated as a risk factor for microalbuminuria (increased albumin in urine) and glomerular filtration rate in diabetic nephropathy [36]. Protein AMBP inhibits calcium oxalate crystallization [37]; since the protein decreases as the body weight increases, it might be a risk factor of formation of stones in individuals with overweight and obesity. Finally, polymeric immunoglobulin receptor has also been identified by capillary electrophoresis coupled to mass spectrometry and proposed as a biomarker in chronic kidney disease [38].

Cluster 3 involves a protein that increases in abundance as the body weight increases. Serum albumin was the only one classified in this cluster. Serum albumin in urine has been used as a biomarker in chronic kidney disease [38].

Cluster 4 classifies proteins that increased in overweight and decreased in obesity. In this cluster, the proteins identified are also involved in different pathologies. Alpha 1-antitrypsin, beta 2-microglobulin, osteopontin and transthyretin are biomarkers for chronical kidney disease [38]. Transthyretin transports thyroxine and retinol. Leucine-rich alpha-2-glycoprotein has been proposed as a potential diagnostic biomarker in acute appendicitis [39]. Identification of urinary proteins offers valuable information not only for kidney or urinary tract related diseases but also a potential picture of the global condition of different organs or tissues in the whole body. Another example of this is the identification of apolipoprotein D (Apo D), a protein with multiple ligands that exert multiple functions in different tissues. It has been associated with various neurological disorders, such as Alzheimer’s disease and Parkinson’s disease or psychiatric disorders including schizophrenia and bipolar disorder [40].

Cluster 5 includes proteins with abundances that increase in women with overweight (W-OW) and decrease in women with obesity (W-OB) and in men with increasing body weight. Interestingly, this group consisted of immunoglobulin heavy chains: immunoglobulin heavy constant gamma 4, immunoglobulin heavy constant gamma 2 and immunoglobulin light chain; immunoglobulin kappa variable 3–20. Immunoglobulins (antibodies) are membrane-bound or secreted glycoproteins produced by B lymphocytes that mediate the effector phase of humoral immunity resulting in elimination of antigens [41]. In obese groups, these proteins decrease in abundance; thus, the immune system might be compromised, and the individuals might be more prone to infections.

Likewise, for proteins in cluster 6, abundance increases in W-OW and decreases in W-OB, whereas in men increases as the body weight increases. Proteins identified in this group included: immunoglobulin lambda constant 2, basement membrane-specific heparan sulphate proteoglycan core protein, which is the major component of the glomerular basement membrane involved in glomerular filtration [42]. Possibly, in the obese group where this protein abundance is decreased, it could indicate compromised glomerular filtration. Inter-alpha-trypsin inhibitor heavy chain H4 is involved in inflammatory responses, and increased abundance of this protein has been associated with early prostate cancer [43].

Finally, proteins in cluster 7 showed a heterogeneous abundance pattern. This cluster includes immune system related proteins: immunoglobulin kappa constant, immunoglobulin heavy constant gamma 1 and mannan-binding lectin serine protease 2, a serum protease that plays an important role in the activation of the complement system via mannose-binding lectin [44]. Hemoglobin subunit beta is normally absent in urine. However, its presence is frequent in urine samples where it may indicate conditions such as nephritis, kidney cancer, malaria and hemolytic anemia; it may also appear in urine in response to strenuous exercise [31]. Uromodulin the most abundant protein excreted in urine under physiological conditions [45]. That this protein was identified in all urine samples supports the accuracy of the method. Furthermore, increased abundance of this protein has been associated with chronic kidney disease [37], suggesting that the level of uromodulin in urine could represent a critical biomarker for kidney function. The last three proteins identified have been associated with different pathologies: prosaposin is a glycoprotein with multiple functions in the body including synthesis and transport of glycosphingolipid and it has been associated with urinary tract infections; cathepsin D is a lysosomal protease and is a urinary marker in renal cell carcinoma [46]; lysosomal alpha-glucosidase is essential for the degradation of glycogen in lysosomes and it has been used as a biomarker in Pompe’s disease, an autosomal recessive disorder characterized by the accumulation of glycogen in all tissues [47]. As discussed before, all proteins identified in pooled urine samples from men or women with normal weight, overweight and obesity are involved in different pathologies from several tissues; however, most of them are urinary tract related. In this sense, Mahyar et al. [48] found a relationship between overweight/obesity and urinary tract infections. Therefore, overweight and obesity might play a role in the pathogenesis of urinary tract alterations.

## 3. Subjects and Methods

### 3.1. Subjects

In this study individuals with normal weight (control), with overweight and with obesity (OB) were selected. Each condition was divided into two groups by gender, for a total of six groups.

Five adult women with normal weight (W-NW) and a mean age at recruitment of 35.2 ± 8.9 years (range 22–47) and a mean BMI of 22.6 ± 0.4.Five adult men with normal weight (M-NW) and a mean age at recruitment of 36.2 ± 11.8 years (range 24–52) and a mean BMI of 22.8 ± 1.1Five adult women with overweight (W-OW) and a mean age at recruitment of 36.8 ± 10.6 years (range 24–51) and a mean BMI of 27.5 ± 1.8.Five adult men with overweight (M-OW) and a mean age at recruitment of 31.6 ± 7.6 years (range 21–41) and a mean BMI of 29 ± 3.8Five adult women with obesity (W-OB) and a mean age at recruitment of 35.8 ± 8.3 years (range 28–46) and a mean BMI of 34.7 ± 2.4.Five adult men with obesity (M-OB) and a mean age at recruitment of 27.6 ± 2.9 years (range 24–32) and a mean BMI of 37.2 ± 4.3.

### 3.2. Sample Collection and Processing

All samples were collected as a midstream portion of the second morning urine (~60 mL); samples were transferred on ice to the lab and processed according to Thomas et al. [5] with slight modifications. Briefly, 45 mL of each urine sample was transferred to a 50 mL conical tube and centrifuged at 1500× *g* for 12 min (Sorvall Lynx 4000, Thermo Scientific, San Jose, CA, USA) to pellet cells. Fifteen milliliters of supernatant was then spun at 10,000× *g* for 12 min and supernatants were transferred to new tubes. Ten milliliters of urine sample of five individuals per group were pooled in a 50 mL conical tube, aliquoted and stored at −40 °C until analysis.

Human urine sample from one healthy donor was selected for FCSNP test (proof of concept) using single and two-dimensional sodium dodecyl sulfate-polyacrylamide gel electrophoresis (2D SDS-PAGE). Additionally, pooled urine from each group of individuals (as indicated before) were selected for MS analysis.

### 3.3. Synthesis and Characterization of Core-Shell Silica Nanoparticles

Nanoparticles were synthesized and characterized as indicated previously [2].

### 3.4. Evaluation of the FCSNP in the Capture of LMW Proteins and Exclusion of HMW in Urine

One milligram of FCSNP was suspended in 150 µL of 0.1% Span 80 [in 20 mM phosphate buffer (PB)] and incubated for 60 min with 150 µL of urine from a healthy donor. After incubation, particles were washed three times with 500 µL of 20 mM PB pH 7.7 and consecutively eluted three times (each with 500 µL) with 50% (*v*/*v*) isopropanol, followed by 50% (*v*/*v*) methanol and then with 70% (*v*/*v*) acetonitrile + 20% (*v*/*v*) ammonium hydroxide (ACN+NH_4_OH). After every wash and elution, nanoparticles were separated from the supernatant by centrifugation at 5585× *g* for 5 min (Costar minicentrifuge, Cambridge scientific, Watertown, MA, USA). Respective supernatants were collected, pooled in a same vial and then dried by centrifugation in vacuum (Centrivap, Labconco, MO, USA). Protein concentration was estimated by Bradford assay [49], using bovine serum albumin (BSA) as the standard. All samples were analyzed by SDS-PAGE [50] and stained with silver [51].

Samples of complete urine (not incubated with FCSNP) and ACN+NH_4_OH elutions product of the incubation of FCSNP with urine from a healthy individual were also prepared for 2D SDS-PAGE separation. For this analysis, 12 mL of complete urine was filtered and washed five times with ultrapure water to remove salts using a MWCO 3 kDa cut-off Centricon (Merck Millipore, Darmstadt, Germany) following manufacturer’s instructions. Meanwhile, to evaluate nanoparticles effectiveness on the capture of LMW proteins, 10 mg of FCSNP was incubated with 10 mL of urine for 1 h. Then, proteins were washed and eluted from the nanoparticles as previously described. All samples were dried by centrifugation under vacuum and then suspended in rehydration buffer (7 M urea, 2 M thiourea, 2% CHAPS, 40 mM dithiothreitol (DTT) and 0.002% bromophenol blue) and mixed by vortexing for 3 min. After vortex, samples were centrifugated at 13,000 rpm for 17 min at 4 °C (Sorvall Lynx 4000, Thermo Scientific, San Jose, CA, USA). Supernatant was precipitated with acetone + 15% Trichloroacetic acid at −20 °C overnight. Samples were then centrifuged at the same conditions, and resultant protein pellets were washed three times with cold acetone, dried and suspended in rehydration buffer.

For 2-DE analysis, isoelectric focusing (IEF) was carried out onto 13 cm IPG linear gradient strips pH 3–10 from GE Healthcare (Piscataway, NJ, USA). Strips were rehydrated with either 100 µg of urinary proteins (not incubated with FCSNP) or 32 µg of urinary proteins eluted with ACN + NH_4_OH (product of the incubation with the FCSNP). IEF was carried out at 20 °C in an Ethan IPGphor 3 IEF System (GE Healthcare, Piscataway, NJ, USA) at 50 mA per strip. The voltage gradient program performed consisted of four steps as follows: (1) 500 V gradient for 1 h, (2) 1000 V gradient for 1 h, (3) 8000 V gradient for 2.5 h and (4) constant 8000 V for 30 min. After IEF, strips were placed onto equilibration buffer (50 mM Tris buffer, 6 M Urea, 30% glycerol, 2% SDS, 0.002% bromophenol blue) plus 1% DTT under agitation for 15 min. Equilibration buffer was discarded and strips were then transferred to equilibration buffer plus 2.5% iodoacetamide for 15 min under agitation. Strips were carefully washed with ultrapure water and then placed onto polyacrylamide gels (13.5%). After electrophoresis, gels were stained with silver [51].

### 3.5. Protein Digestion and Analysis by Liquid Chromatography-Tandem Mass Spectrometry (LC-MS/MS) of Pooled Samples

Ten milligrams of FCSNP was suspended in 1.5 mL of 0.1% span 80 and incubated for 1 h with 10 mL of pooled urine samples from each group (Section 2.1). After incubation, samples were centrifuged at 10,000× *g* for 7 min (Sorvall Lynx 4000, Thermo Scientific, San Jose, CA, USA) and respective supernatants (non-binding fraction) were transferred to new conical tubes and stored at −40 °C. Proteins retained by FCSNP were washed and then eluted in a modified process as follows: three washes with 1.5 mL 20 mM PB followed by three elutions with 1.5 mL ACN+NH_4_OH, performing centrifugation in between washes or elutions as indicated before. Respective supernatants were partially dried, then combined in a same vial and completely dried by centrifugation in vacuum (Vacufuge Plus, Eppendorf, Hamburg, Germany). Complete process was performed in triplicate. Samples were stored at −40 °C until further digestion and analysis by MS.

Eluted proteins with ACN+NH_4_OH were selected for LC-MS/MS analysis. Estimated 30 µg of total protein from each sample were reconstituted in 1 M urea, reduced with 10 mM DTT, alkylated with 50 mM iodoacetamide and digested with sequencing grade trypsin (Promega, Madison, WI, USA) overnight at 37 °C. Tryptic peptides were separated in a reverse phase ultra-performance liquid chromatography using a 1290 Infinity LC system (Agilent Technologies, Santa Clara, CA, USA). Analytical column used was a ZORBAX SB-C18 (100 × 2.1 mm, 1.8 µm, Agilent Technologies, Santa Clara, CA, USA) equilibrated with 1% ACN and 0.1% formic acid (FA) maintained at 35 °C [22,52] with a flow rate of 400 µL/min.

Peptides were then separated with the following chromatographic conditions: 1 min with 99% of solution A (H2O with 0.1% FA) and 1% of solution B (ACN with 0.1% FA), followed by two linear gradients until reaching 40% of B in 60 min and then 90% of B in 20 min (from 60 to 80 min) and a final gradient of 1 min until reaching 1% of B, with an equilibrium period with 1% of B for 5 min between runs. Peptides eluted from the column were ionized by electrospray with a Dual AJS ESI ionization source applying 3.5 kV, and ions were then analyzed by MS/MS in data-dependent acquisition mode in a 6530 Accurate-Mass Quadrupole Time-of-Flight (Q-ToF) LC/MS System (Agilent Technologies, Santa Clara, CA, USA) and the conditions reported by Morales-Amparano [52] with a brief change: data were acquired in MS mode at 4 spectra/s scan rate, whereas for MS/MS scan the rate was set at 1 spectrum/s and a maximum of 5 precursors per MS cycle were selected for further peptide fragmentation by collision-induced dissociation.

### 3.6. Protein Identification and Differential Abundance Cluster Representation (Heat Map)

MS data were searched against the *Homo sapiens* subset of the Swiss-Prot protein database (20,206 sequences, June 2017). Protein identification was performed using the Spectrum Mill MS Proteomics Workbench server (Agilent Technologies, Santa Clara, CA, USA). Specific protease selected was trypsin, allowing one missed cleavage. Mass error tolerance for parent and fragment ions was set at 20 ppm and 0.1 Da, respectively. Carbamidomethyl cysteine was selected as fixed modification, while methionine oxidation was selected as variable modification. Individual ions scores ≥ 9 and scored peak intensity (SPI) ≥ 60 were considered good matches, whereas protein score ≥ 25 and at least two peptides were necessary for a confident protein identification.

Differential abundance patterns of identified proteins were performed on the log transformed abundance ratios in men or women with overweight or obesity compared to controls (normal weight), from which seven clusters were established. The web server heatmapper (http://www.heatmapper.ca accessed on September 2020) was used to generate the heat map representation of the results. Proteins were considered as differentially accumulated when their values had a fold change ≥2.

## 4. Conclusions

The combined approach of nanotechnology and proteomics is a novel strategy for rapid evaluation of changes occurring in biological systems. The nanoproteomic analysis used here could serve as an important tool in defining biomarkers.

FCSNP enabled, in a single step, the capture of LMW proteins (potential biomarkers), exclusion of the HMW proteins and the simultaneous removal of salts from urine samples; this FCSNP, followed by a proteomic analysis, was clearly able to distinguish differences in protein presence and abundance among individuals of different gender and various body weight conditions. It is important to point out that validation with a larger group of individuals is required.

## Figures and Tables

**Figure 1 molecules-26-01803-f001:**
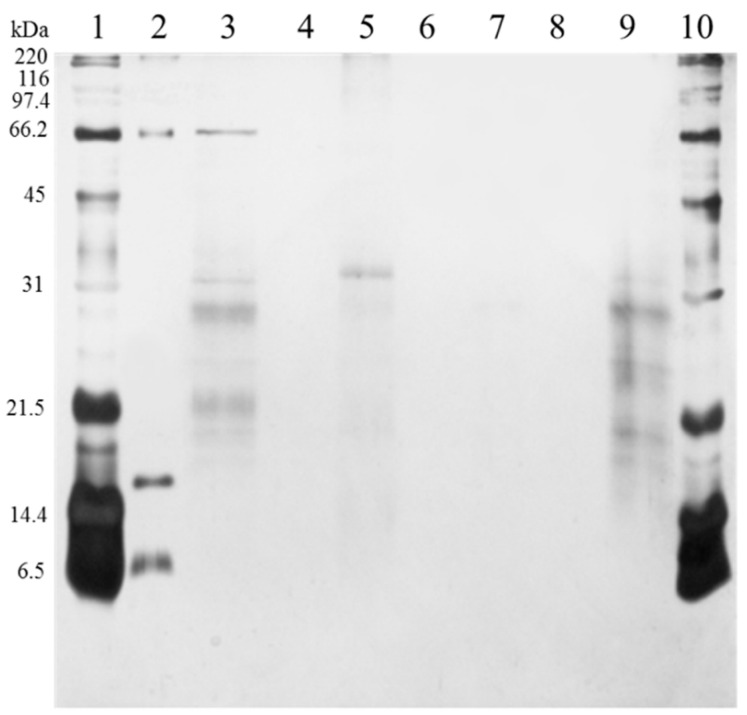
SDS-PAGE analysis of the functionalized core-shell silica nanoparticles (FCSNP) elutions after incubation with urine from a healthy donor for 60 min. Lanes were loaded with 1 µg of protein and arranged as follows. Broad range molecular weight standards (lanes 1 and 10); BSA, 66 kDa, myoglobin, 17 kDa, and aprotinin, 6.5 kDa (lane 2); urine, without passing through FCSNP (lane 3); sample buffer (lanes 4, 6 and 8); elution with 50% isopropanol (lane 5). Simultaneous exclusion of high molecular weight proteins and capture of low molecular weight proteins ~34 kDa and smaller was achieved in elution with 50% methanol, one faded band ~30 kDa (lane 7) and ACN+NH_4_OH (lane 9).

**Figure 2 molecules-26-01803-f002:**
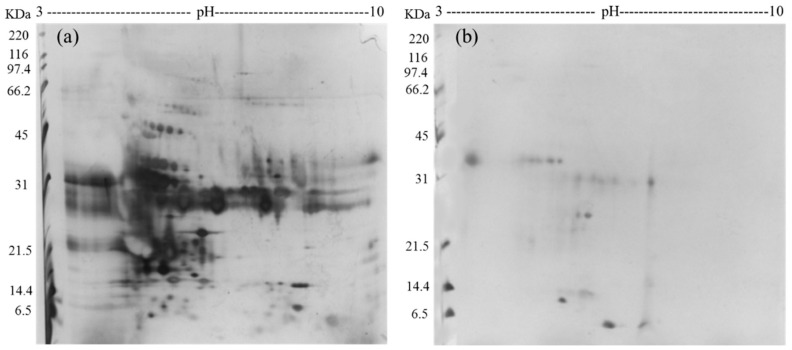
2D SDS-PAGE analysis of the complete urine (from a healthy donor), not incubated with FCSNP (**a**) and ACN+NH_4_OH elution, final product of the incubation of urine (from same healthy donor) with the FCSNP (**b**). In Figure 2a all proteins (high molecular weight (HMW) and low molecular weight (LMW) proteins) in urine are shown. Simultaneous exclusion of HMW and capture of LMW proteins (~34 kDa and smaller) of the FCSNP was achieved in the ACN+NH_4_OH elution (2B).

**Figure 3 molecules-26-01803-f003:**
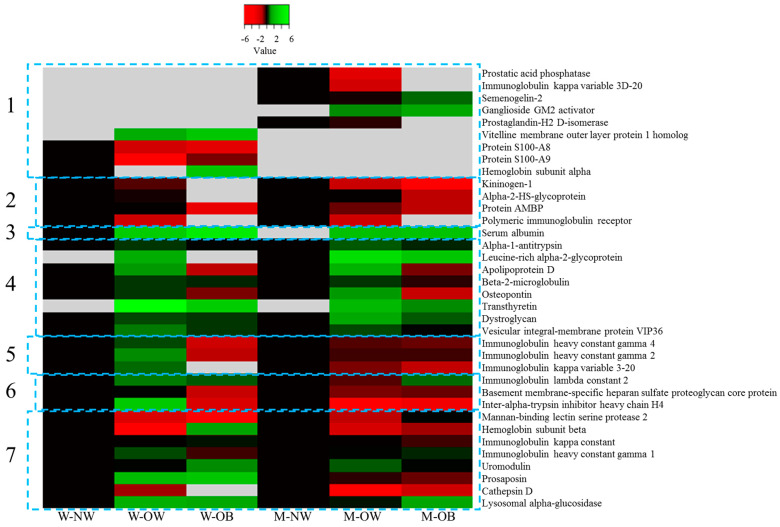
Heat map of the 36 proteins identified by LC-MS/MS. Columns represent groups of women (W-NW) or men (M-NW) with normal weight (controls), overweight (W-OW, M-OW) and obesity (W-OB, M-OB). Rows indicate individual identified proteins. Proteins not detected in any group/condition are indicated in gray. Increased and decreased protein abundance in the different groups compared to respective normal weight conditions are indicated in green or red, respectively. Protein names are indicated on the right side. Dashed blue boxes indicate protein clusters with similar abundance change patterns.

**Figure 4 molecules-26-01803-f004:**
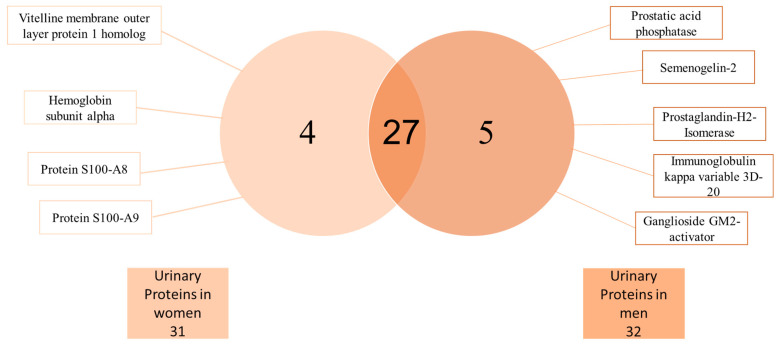
Basic Venn diagram. A total of 36 urine proteins were identified by LC-MS/MS analysis. Four proteins were only found in women and five only in men, whereas 27 proteins were found in both genders.

**Table 1 molecules-26-01803-t001:** Identification of proteins with different abundance patterns in urine pooled samples from groups of men and women with normal weight (control), overweight and obesity.

Protein	Accession Number	PMM (Da) ^a^	Score	PM/Sc % ^b^	W-NW ^c^	Fold Change		M-NW ^f^	Fold Change ^i^	
						W-OW ^d^	W-OB ^e^		M-OW ^g^	M-OB ^h^
**Cluster 1. Protein present/absent in either men or women.**										
Prostatic acid phosphatase	P15309	44,907.6	33.62	2/6.9	X	X	X	√	0.19	X
Immunoglobulin kappa variable 3D-20	A0A0C4DH25	12,628.9	28.35	2/21.5	X	X	X	√	0.27	X
Semenogelin-2	Q02383	65,557.4	29.46	2/4.8	X	X	X	√	0.90	1.68
Ganglioside GM2 activator	P17900	21,294.4	27.96	2/15.5	X	X	X	X	√	√
Prostaglandin-H2 D-isomerase	P41222	21,256.7	25.74	2/17.3	X	X	X	√	0.83	X
Vitelline membrane outer layer protein 1 homolog	Q7Z5L0	22,047.2	49.96	4/33.6	X	√	√	X	X	X
Protein S100-A8	P05109	10,891.4	46.88	3/31.1	√	0.27	0.18	X	X	X
Protein S100-A9	P06702	13,298.8	44.4	3/31.5	√	0.08	0.57	X	X	X
Hemoglobin subunit alpha	P69905	15,314.3	42.55	3/21.1	√	X	√	X	X	X
**Cluster 2. Protein abundance decreases as the body weight increases.**										
Kininogen-1	P01042	73,040.1	116.52	9/16.3	√	0.68	X	√	0.34	0.03
Alpha-2-HS-glycoprotein	P02765	40,122.7	76.81	5/20.7	√	0.85	X	√	0.95	0.42
Protein AMBP	P02760	39,911.7	52.14	3/10.5	√	0.93	0.17	√	0.64	0.43
Polymeric immunoglobulin receptor	P01833	84,480.2	51.28	3/6.5	√	0.30	X	√	0.29	X
**Cluster 3. Protein abundance increases as the body weight increases.**										
Serum albumin	P02768	71,362.3	84.63	6/9.8	√	6.79	22.23	X	√	√
**Cluster 4. Protein abundance increases in overweight (OW) and decreases in obesity (OB).**										
Alpha-1-antitrypsin	P01009	46,906.8	120.34	7/22	√	1.32	1.03	√	1.64	1.26
Leucine-rich alpha-2-glycoprotein	P02750	38,405.4	68.27	5/24.4	X	√	X	√	12.64	5.54
Apolipoprotein D	P05090	21,560.4	67.97	5/24.3	√	2.12	0.43	√	3.42	0.60
Beta-2-microglobulin	P61769	13,828.4	49.73	3/35.2	√	1.38	1.20	√	1.31	0.78
Osteopontin	P10451	35,593.3	42.73	3/14.9	√	1.35	0.60	√	2.19	0.35
Transthyretin	P02766	16,000.8	31.48	2/244	X	√	√	X	√	√
Dystroglycan	Q14118	97,782	27.58	2/2.6	√	1.44	1.33	√	2.71	1.65
Vesicular integral-membrane protein VIP36	Q12907	40,570.3	88.09	6/18.8	√	1.89	1.33	√	1.44	1.00
**Cluster 5. Protein abundance increases in W-OW and decreases in W-OB. Decreases as body weight increases in men.**										
Immunoglobulin heavy constant gamma 4	P01861	36,453.4	69.58	5/22.6	√	1.46	0.29	√	0.68	0.63
Immunoglobulin heavy constant gamma 2	P01859	36,527.6	47.65	4/15.9	√	2.11	0.42	√	0.73	0.72
Immunoglobulin kappa variable 3–20	P01619	12,671	30.84	2/21.5	√	1.66	X	√	0.56	0.38
**Cluster 6. Protein abundance increases in W-OW and decreases in W-OB. Increases as body weight increases in men.**										
Immunoglobulin lambda constant 2	P0DOY2	11,464.5	72.9	5/74.5	√	1.88	1.65	√	0.71	1.68
Basement membrane-specific heparan sulphate proteoglycan core protein	P98160	479,547.8	70.1	5/1.6	√	1.06	0.35	√	0.59	0.65
Inter-alpha-trypsin inhibitor heavy chain H4	Q14624	103,583.8	49.44	3/6.6	√	6.05	0.22	√	0.04	0.13
**Cluster 7. Proteins with heterogeneous abundance patterns.**										
Mannan-binding lectin serine protease 2	O00187	77,241.4	107.62	710.4	√	0.23	0.06	√	0.43	1.02
Hemoglobin subunit beta	P68871	16,112.2	115.05	6/46.9	√	0.09	2.34	√	0.24	0.51
Immunoglobulin kappa constant	P01834	11,936	95.38	679.4	√	0.96	1.12	√	0.92	0.77
Immunoglobulin heavy constant gamma 1	P01857	36,618.7	85.98	6/26.9	√	1.50	0.72	√	1.05	1.24
Uromodulin	P07911	72,498.3	74.37	6/9	√	1.02	2.07	√	1.55	1.00
Prosaposin	P07602	59,937.4	35.85	2/4.7	√	4.30	5.47	√	0.80	0.62
Cathepsin D	P07339	45,064.9	29.75	2/5.5	√	0.51	X	√	0.07	0.31
Lysosomal alpha-glucosidase	P10253	106,177.6	31.4	2/3.5	√	3.18	2.69	√	1.17	2.41

^a^ Protein molecular mass (Da). ^b^ Number of peptides matched/sequence coverage percentage. ^c^ Women with normal weight (control); presence or absence of proteins in normal weight conditions (controls) are indicated with √ or X, respectively. ^d^ women with overweight; ^e^ women with obesity. ^f^ men with normal weight (control); ^g^ men with overweight; ^h^ men with obesity. ^i^ fold change is expressed as the ratio of the abundance of proteins in overweight or obesity/normal weight. Each value represents the mean value of three independent measurements. Fold change cannot be accurately calculated due to absence of proteins in either normal weight (control) or overweight or obesity conditions; thus, √ or X were also used to indicate presence or absence of proteins, respectively. Decreased (fold change ≤ 0.5) or increased (fold change ≥ 2) protein abundance are highlighted in light gray or dark gray, respectively.

## Data Availability

Not applicable.

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
