# Peer review of "Nanoproteomic Approach for Isolation and Identification of Potential Biomarkers in Human Urine from Adults with Normal Weight, Overweight and Obesity"

_molecules, 2021, doi:10.3390/molecules26061803_

Round 1

Reviewer 1 Report

The manuscript by Hernandez-Leon et al. describes the use of restricted-access polymer-coated silica particles for the capture of low-molecular weight protein and peptide biomarkers from urine, for proteomics.

This manuscript is the follow-up of their earlier paper where the new adsorption material was introduced with model samples. In the present manuscript the authors now present its use with real samples (male and female urine), and report proteomics data.

I believe the manuscript is original and can be published after minor modifications.

Although this was explained in their earlier paper, could the authors comment again on the rational of using Cibacron blue as bait ligand, and the possible bias introduced by this ligand. This ligand is known from the origins of affinity chromatography as a general group-specific ligand. Is there any alternative ligand that could be used in parallel to broaden the range of recovered molecules?

What about the (low molecular-weight) proteins remaining in the non-retained fraction, relative to those retained? Have they been identified and could they be captured otherwise?

From a technical point of view, have the about authors thought about the use of magnetic NP-silica composites for easier handling?

Author Response

Response to Reviewer 1 Comments

Point 1. I believe the manuscript is original and can be published after minor modifications. Although this was explained in their earlier paper, could the authors comment again on the rational of using Cibacron blue as bait ligand, and the possible bias introduced by this ligand. This ligand is known from the origins of affinity chromatography as a general group-specific ligand.

Response: Thank you for reviewing this article and for providing these valuable observations. Modifications were made and can be found in the manuscript (introduction section) highlighted in yellow.

Point 2. Is there any alternative ligand that could be used in parallel to broaden the range of recovered molecules? What about the (low molecular-weight) proteins remaining in the non-retained fraction, relative to those retained? Have they been identified and could they be captured otherwise?

Response: Since silica can be chemically modified with different compounds, there are many different ligands that can be used as an alternative. In fact, we have already modified the core of the silica nanoparticles with iminodiacetic acid for the capture of His-tagged recombinant proteins (Hernandez-Leon et al., 2019). In this sense, yes, chemically modified core-shell silica nanoparticles could be used in parallel to broaden the range of recovered molecules. However, in this study, we aimed to use these nanoparticles exclusively since they have been tested before with model proteins and their performance with more complex biological samples had not been evaluated. Proteins in the non-retained fraction have not been identified, and although that is out of the scope of this paper, it would be very interesting to modify the nanoparticles with molecular baits with different characteristics (hydrophilic, for example) for the capture of those proteins.

Point 3. From a technical point of view, have the about authors thought about the use of magnetic NP-silica composites for easier handling?

Response: The aim of the study was to evaluate the performance of previously synthesized core-shell silica nanoparticles (without modifications) with a more complex biological sample. However, we do not discard the option of synthesizing magnetic nanoparticles in the future since they are very easy to handle, although additional steps in the synthesis are required. Based on different important works in this matter such as Sahu et al., 2010, Salimi et al., 2017 and Zhou et al., 2018, just to mention a few, we hypothesize that it is possible, eventually, to synthesize core-shell magnetic silica nanoparticles for the capture of proteins with different characteristics.

Manuscript with respective modifications (indicated in yellow) is attached. 

Reviewer 2 Report

The themathic of the article is very important, the introduction is ample and the purpose of the study is clearly specified, the methodology is adequate, modern, complex investigations are used, the results are exceptional and very usefull in clinical practice. Please further improve the graphical part and very clerarly promothe the main novel markers you have identified because they are of extreme importance. Please in the discussion section discuss wether pollutants can infulenece certain human proteins an this modified protein can be found as markers of obseity. Please reffer to the fallowing papers

Saramasan C, Identification, Communication And Management Of Risks Relating To Drinking Water Pollution In Bihor County, Environmental Engineering and Management Journal, November/December 2008, Vol.7, No.6, 769-77

Yusa V, Ye X, Calafat AM. Methods for the determination of biomarkers of exposure to emerging pollutants in human specimens. Trends Analyt Chem. 2012;38:129-142. doi:10.1016/j.trac.2012.05.004

Author Response

Response to Reviewer 2 Comments

Point 1. The themathic of the article is very important, the introduction is ample and the purpose of the study is clearly specified, the methodology is adequate, modern, complex investigations are used, the results are exceptional and very usefull in clinical practice. Please further improve the graphical part and very clerarly promothe the main novel markers you have identified because they are of extreme importance. Please in the discussion section discuss wether pollutants can infulenece certain human proteins an this modified protein can be found as markers of obseity. Please reffer to the fallowing papers

Saramasan C, Identification, Communication And Management Of Risks Relating To Drinking Water Pollution In Bihor County, Environmental Engineering and Management Journal, November/December 2008, Vol.7, No.6, 769-77

Yusa V, Ye X, Calafat AM. Methods for the determination of biomarkers of exposure to emerging pollutants in human specimens. Trends Analyt Chem. 2012;38:129-142. doi:10.1016/j.trac.2012.05.004

Response: Thank you for reviewing this article and for providing these valuable observations. Information about water pollutants and references have been included in the manuscript (highlighted in yellow) attached.
